# Detailing the Ten Main Professional Roles of a Pharmacist to Provide the Scope of Professional Functions

**Yuliia Kremin** [1], **Lilia Lesyk** [2] , **Roman Lesyk** [3,*] , **Oksana Levytska** [1] and **Bohdan Hromovyk** [1]

1    Department of Organization and Economics of Pharmacy, Danylo Halytsky Lviv National Medical University, 69 Pekarska, 79010 Lviv, Ukraine
2    Department of Business Economics and Investment, Institute of Economics and Management, Lviv Polytechnic National University, 5 Metropolian Andrey Str., Building 4, 79005 Lviv, Ukraine
3    Department of Pharmaceutical, Organic and Bioorganic Chemistry, Danylo Halytsky Lviv National Medical University, 69 Pekarska, 79010 Lviv, Ukraine
*    Correspondence: dr_r_lesyk@org.lviv.net; Tel.: +38-(032)-275-59-66

**Abstract:** As members of a public trust profession, pharmacists are the most accessible medical team members. Therefore, every pharmacist must know the scope of their professional roles (PR) and professional functions (PF). The study aimed to detail the major PR into a pooled set of PF. The research materials were the provisions of the World Health Organization, the International Pharmaceutical Federation, and scientific works on the PR of pharmacists. Methods of critical analysis, concretization, functional decomposition, and scientific generalization were used. As a result of detailing the 10 main PR according to the "ten-star pharmacist" concept for each, a combined set of partial PFs of the pharmacist was obtained. The decomposition takes into account the principle of complexity limitation, which allowed three to six partial PF for the respective PR to be obtained, namely: three PFs for a life-long-learner, five PFs for a caregiver, a decision-maker, a teacher, a leader, a researcher, an entrepreneur, and an agent of positive change, six PFs for a communicator and a manager. Thus, due to the decomposition of each of the 10 main PR of the pharmacist into three or six corresponding partial PFs, we received a multifunctional verbal model of difficult to organize, professional activities, which is identified by a total of 50 PFs. The importance of using this model in formulating professional competencies and learning outcomes of educational programs for pharmacists is emphasized.

**Keywords:** "ten-star pharmacist" concept; professional role; decomposition; professional function; professional education programs



## 1. Introduction

As members of a public trust profession [1], pharmacists play an essential role in health care, as their work is directly linked to global goals and challenges. Accordingly, the key professional roles (PR) of the pharmacist are to search for, prepare, compound, supply, store, sell, and organize the disposal or destruction of drugs, ensuring and controlling the safe and effective use of drugs to obtain the desired therapeutic effect and to minimize drug-related problems. Given the multifunctionality of the key pharmacist's PR, in 1997, the Advisory Group of the World Health Organization proposed the "seven-star pharmacist" concept, which was approved by the International Pharmaceutical Federation in 2000, identifying seven major PR of the pharmacist: a caregiver, a decision-maker, a communicator, a manager, a life-long learner, a teacher, and a leader [2,3]. Further development of pharmacists' multifunctionality led to expanding their main PR [4–6] by adding new professional components—a researcher, an entrepreneur, and an agent of positive change. So, as of now, the "ten-star pharmacist" concept is spreading in the world. This concept is gaining scientific justification and content at the level of researchers from different countries [7–11]. PR of the pharmacist are relatively stable patterns of thoughts, experiences, and actions

that are driven by and relevant to the demands of the profession. However, an analysis of the literature showed that the 10 main PR of the pharmacist received only a general interpretation without their detail [8,12–18]. Therefore, the study aimed to decompose each of the major PR of pharmacists into a pooled set of professional functions (PF).

## 2. Materials and Methods

The study was based on the provisions of the World Health Organization, the International Pharmaceutical Federation, and scientific publications on the PR of the pharmacist. Inductive content analysis was used to study informational materials [19], a qualitative analysis of text arrays aimed at further meaningful interpretation concerning the main PR and partial PF. Functional decomposition is used to clarify the PF, an analysis method that breaks down a complex process into its individual elements [20]. In turn, scientific generalization [21] made it possible to correlate partial PF by similar properties or connections to the main PR. A simplified interpretation and visual representation of the decomposition of the key PR pharmaceutical specialist was carried out using the modeling method [22].

## 3. Results

As a result of the decomposition of 10 PR of the pharmacist, we obtained a combined set of partial PFs concerning their main tasks (directions) and activities in Figure 1. The detailing took into account the principle of complexity limitation [23], the essence of which is that the obtained partial PF of the pharmacist should be legible and easy to read, and their number within the main PR should be at least three and at most six.

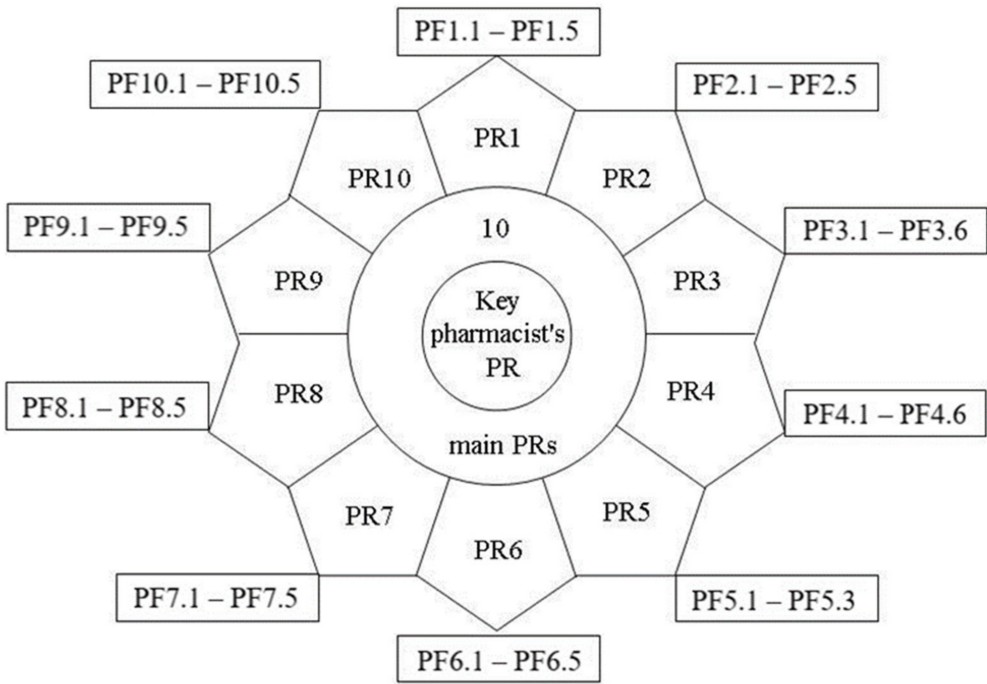

**Figure 1.** Decomposition of the main PR of the pharmacist.

Adherence to this principle has led to the pharmacist's detailed PF of the main PR being well structured, understandable, and easy to analyze [24,25].

### 3.1. Caregiver

According to the results, the pharmacist, as a caregiver (major PR1), should provide patients with the highest quality pharmaceutical services and consider their practice in the context of other doctors and the health care system. In this case, they must implement five partial PF, namely:

- assess the state of health, the level of his medical literacy, and physiological needs for the quality of pharmaceutical care (PF1.1);
- to ensure the implementation of possible diagnostic measures (measurement of blood pressure, heart rate, blood sugar, etc.) in a pharmaceutical institution (PF1.2);
- provide quality pharmaceutical care through a set of its characteristics (appropriate therapeutic concordance, patient-oriented results, economic and physical accessibility, rationality, safety, effectiveness, timeliness, absence (minimization) of drug-related mistakes, continuity) (PF1.3);
- have the skills of telepharmacy, i.e., remote (distant) provision of quality pharmaceutical care with the assistance of a range of organizational and financial measures, information and telecommunications technologies, and infrastructure (PF1.4);
- take professional responsibility for the quality of pharmaceutical care, including in emergencies, pandemics, and peacetime and wartime emergencies, as well as for reducing environmental risk throughout the drug supply circulation chain (PF1.5).

### 3.2. Decision Maker

According to the basic PR2 (a decision-maker), a pharmacist must be able to choose a specific course of action from a variety of alternatives, monitor compliance, make adjustments as necessary, and evaluate the results of a decision. PR2 is also detailed in five partial PF:

- rational, cost-effective procurement and efficient, safe, and economical use of all resources in the pharmaceutical organization (medicines, medical devices, active pharmaceutical ingredients, excipients, etc.) (PF2.1);
- decision-making on the proper organization of the pharmaceutical organization and participation in the production (manufactured) of medicines in the conditions of pharmaceutical enterprises and pharmacies (PF2.2);
- determination of factors influencing the processes of absorption, distribution, deposition, metabolism, and excretion of drugs due to the condition, features of the human body, and physicochemical properties of drugs (PF2.3);
- the adoption of safe and logical decisions based on proper evaluation of evidence-based medicine and pharmacy data and professional judgments, including the choice of a rational type of pharmaceutical care, which minimizes the risk of pharmacotherapy for humans and promotes the provision of quality pharmaceutical care (PF2.4);
- influence on the implementation of the national medical policy, through active participation in the activities of pharmaceutical public and professional organizations, as a consequence of promoting the improvement of the regulatory framework of the pharmaceutical sector of the health sector (PF2.5).

### 3.3. Communicator

According to the PR3 (a communicator), the pharmacist should serve as a bridge between the patient and the doctor and provide the public with information about health and medicines. We decompose PR3 into six partial PF, namely the pharmacist must:

- possess and apply in practice verbal and nonverbal skills of interpersonal communication and fundamental principles of pharmaceutical ethics and deontology (PF3.1);
- establish regular bilateral and multilateral communications in the process of pharmaceutical care aimed at doctors, medical staff, patients, or their caregivers (PF3.2);
- show tolerance, loyalty, respect, compassion, and understanding to the patient regardless of nationality, political and religious beliefs, property status, gender, age, and social status (PF3.3);
- keep confidential information about the condition, health, and diagnosis of the patient, which is a professional secret, except as provided by law (PF3.4);
- ensure that information for patients, doctors, other health professionals, and the public is based on evidence-based medicine and is objective, effective, unbiased, understandable, non-promotional, accurate, and adequate (PF3.5);

– to carry out information work among the population to prevent the use of potentially harmful substances for recreational purposes, to prevent substance abuse, prevent common diseases, prevent dangerous infectious, viral and parasitic diseases, as well as to promote timely detection and maintenance of treatment for these diseases, their medical and biological characteristics and microbiological features (PF3.6).

### 3.4. Manager

The major PR4 of the pharmacist as a manager includes the ability, based on professional knowledge of organizational management, to effectively influence employees, set up specific work, and positively perceive leadership from other professionals. They are characterized by six partial PF, namely:

– to carry out planning, i.e., determine the prospects and future state (landmark of future activities) of the pharmaceutical organization (PF4.1);
– to be able to organize, i.e., create an adequate structure of the pharmaceutical organization by dividing it into units according to goals and strategies, and to establish the relationship of powers of higher and lower levels of government and opportunities for division and coordination of tasks (PF4.2);
– to motivate, i.e., to ensure the process of motivating yourself and other members of the pharmaceutical organization to activities aimed at achieving goals, both personal and for the whole organization (PF4.3);
– to control, i.e., determine how to make correct management decisions, and determine, as well, the need to make certain adjustments (PF4.4);
– to regulate, i.e., eliminate deviations, failures, shortcomings, etc., in the pharmaceutical organization by developing and implementing appropriate measures (PF4.5);
– to adequately respond to managerial influence in the vertical and horizontal division of managerial activity (PF4.6).

### 3.5. Life-Long Learner

Regarding the fifth main PR5—being a life-long-learner, according to which pharmacists must begin with the stage of specialist training and continue training throughout their professional careers—it is detailed into three partial PF, according to which pharmacists must:

– identify their learning needs and be individually responsible for the systematic acquisition, maintenance, development, and expansion of program competencies throughout their professional activities (PF5.1);
– be active participants in the two stages of continuing pharmaceutical education (specialist training and continuing professional development) (PF5.2);
– use three forms (formal, non-technical, and informal) of implementation and two forms (institutionalized and non-institutionalized) of education (PF5.3).

### 3.6. Teacher

In turn, the basic PR6 (a teacher), where pharmacists help in the education and training of future generations of their colleagues and informing society, is divided into five partial PF, i.e., pharmacists must:

– transfer to the new generation of pharmaceutical specialists their knowledge, skills, and abilities, as well as encourage them to demonstrate independent cognitive activity, including through mentoring (PF6.1);
– be able to properly organize the educational process in the pharmaceutical organization, which would be convenient for the new generation of pharmaceuticals, and would not interfere with the work of employees of this organization and their patients (PF6.2);
– participate in educating patients on issues related to the protection and promotion of health, medical literacy, and effectively promote a healthy lifestyle using scientifically sound methods (PF6.3);

- provide training to empower patients and their communities in self-care in health care, ultimately to help optimize the use of resources and costs for health care and to improve outcomes for patients and health care systems (PF6.4);
- teach patients the rules of counteracting the distribution of counterfeit medicines and the disposal of pharmaceutical waste as an element of pharmaceutical practice (PF6.5).

### 3.7. Leader

The basic PR7 characterizes a pharmacist as a leader, i.e., a specialist who, due to personal qualities, inspires colleagues to show their abilities in ensuring the well-being of patients and society and, in the case of leadership positions, understands leadership mechanisms and can cope with the burden of power. Furthermore, according to the content of this PR, pharmacists implement five partial PF, i.e., they must:

- have organizational skills, be able to gain trust and persuade members of the inter-disciplinary medical team and the staff of the pharmaceutical organization, combine their efforts to achieve the goals of the multidisciplinary team and the pharmaceutical institution (PF7.1);
- establish, maintain and coordinate favorable personal, professional, and socio-psychological relations within the team of the pharmaceutical organization and between team members, doctors, nurses and patients, and their caregivers in compliance with professional ethics, resolve conflicts in professional activities, and also support their colleagues at all stages of the work (PF7.2);
- consciously introduce new, constructive ideas for the functioning and development of the staff of the pharmaceutical organization, including the formulation of new goals and objectives, justifying priorities in the development of the team, and tactical ways and methods of achieving them (PF7.3);
- promote the achievement of the goals of the pharmaceutical organization, high performance, positive changes in colleagues, and be responsible for the results of collective activities (PF7.4);
- promote the profession's prestige, particularly its life position and professional competence, thus serving as a model of behavior for colleagues, members of a multidisciplinary medical team, and patients (PF7.5).

### 3.8. Researcher

According to the main PR8 (a researcher), pharmacists should focus on the search and development of new or improved existing drugs, effectively use the evidence base to provide recommendations for the rational use of drugs in the medical team, contribute to the evidence base in order to improve the treatment of patients, etc. Pharmacists act as researchers as a result of the implementation of five partial PF, during which they must:

- understand the concept of construction and facilitate various types of research, make hypotheses, build theories, and obtain facts to implement the results of pharmaceutical research in the evidence base and practice (PF8.1);
- search for and develop new or improve existing drugs, identify the advantages and disadvantages (unknown in practice, undesirable, threatening side effects) of drugs of different pharmacological groups, and recognize the need to apply the principles of pharmacovigilance (PF8.2);
- develop new methods of quality control of medicines, including active pharmaceutical ingredients, medicinal plant raw materials, and excipients, predict and determine the impact of environmental factors on the quality of medicines and consumer characteristics of related products following their physicochemical properties, and recognize that these will be affected by the individual size, weight, gender, and age of consumers (PF8.3);
- based on audit and evaluation of provided pharmaceutical services, develop and implement new approaches to the implementation of professional pharmaceutical activities, in particular through the quality management system in the pharmaceutical organization, and ensure their effective updating (PF8.4);

–　effectively use the modern system of evidence on medicines to provide recommendations for the rational use of medicines in a multidisciplinary medical team, and contribute to the evidence base by accumulating an array of evidence-based scientific and pharmaceutical information, disseminating it in multidisciplinary cooperation to improve the treatment of patients (PF8.5).

*3.9. Entrepreneur*

The major PR, PR9 (an entrepreneur), focuses on independent, systematic, proactive, and risky activities related to the production of medicines and pharmaceutical services to promote the welfare of society for profit or personal income, and it involves innovation and implementation of innovations. This basic PR is detailed by us into five partial PF, according to which the pharmacist must:

–　take the initiative to combine financial, production, material, raw materials, human, informational, intellectual, and other resources in the process of circulation of medicines (PF9.1);
–　generate innovations (novelty), master new medicines, technologies, and forms of pharmaceutical business organization, search for new markets, means of meeting consumer needs, the transition from traditional to new forms of management (PF9.2);
–　risk one's property, ownership, invested funds, and one's labor, time, and business reputation in the course of entrepreneurial activity (PF9.3);
–　make decisions at the stages of circulation of medicines, which are aimed at achieving the success of the pharmaceutical organization, but do not guarantee it due to the uncertainty and variability of the economic situation (PF9.4);
–　implement the trinity of the pharmaceutical business to obtain its economic benefits and ensure the economic well-being of its employees and the maximum satisfaction of consumer needs (PF9.5).

*3.10. Agent of Positive Change*

As an agent of positive change (major PR10), the pharmacist should facilitate the process of change in pharmaceutical practice to improve patient care, the quality of pharmaceutical services, and multidisciplinary cooperation. This final one of the PR is described as five partial PF, namely:

–　in-depth analysis of the state of development of pharmaceutical practice, identification of strategic and common priorities in ensuring the quality of pharmaceutical services, participation in the creation or improvement of national professional responsibilities, guidelines, and legislation that guide the pharmaceutical community to these priorities (PF10.1);
–　education of the pharmaceutical community on essential issues in the development of pharmaceutical practice, advising on difficult issues to which the answers are not obvious, cultivating a spirit of expertise, and involving other experts in creating common knowledge bases to improve patient care (PF10.2);
–　involvement of members of the pharmaceutical community in projects aimed at improving patient care, and quality of pharmaceutical services, using modern methods of online and offline channels (PF10.3);
–　active adaptation to new hybrid challenges of modern times, acting socially responsibly, in particular through participation in various public and social events guiding the transformation of human behavior to actively overcome complex threats (PF10.4);
–　motivation to work in public and professional pharmaceutical organizations on a volunteer basis (PF10.5).

## 4. Discussion

According to the results of our analysis, the ten main PR of a pharmacist were detailed, and a multifunctional model of difficult to organize, professional activities of the pharmacist in the form of 50 partial PF was obtained, namely:

- three partial PF are characteristic of a life-long-learner;
- five partial PF—for a caregiver, a decision-maker, a teacher, a leader, a researcher, an entrepreneur, and an agent of positive change;
- six partial PF—for a communicator and a manager.

However, according to previous research, only 38.4% of surveyed pharmacists, 34.6% of students—future Master of Pharmacy graduates, and 10.0% of visitors to pharmacies in Ukraine correctly identified the components of the "ten-star pharmacist" concept [9]. On the one hand, the unsatisfactory level of awareness of students and pharmacists with the components of the above concept may be the reason for providing inadequate pharmaceutical care. On the other hand, the lack of understanding of patients of the importance and multifunctionality of pharmacist's work is the reason for their non-perception of pharmacies as health care facilities, as well as the transformation judgment of society, and often pharmacists themselves, relating the pharmacist's PR to a seller's work.

The patient's perception of the services provided by the pharmacist is critical, as it will serve the future perspective, which can be used to determine the vector of PR development, and thus the pharmacist's PF, which will emphasize the importance of the pharmacist and improve the quality of pharmaceutical care [26,27].

The results of another study on educational and professional programs (EPP) for training Master of Pharmacy students, at twenty-two universities in Ukraine, show significant differences between these EPPs. In particular, 12 of the 15 general and only 16 of the 35 professional competencies are common in these EPPs. At the same time, they do not sufficiently reflect the PRs of a life-long-learner, a decision-maker, and a teacher, i.e., one-third of the current PRs, which are part of the "ten-star pharmacist" concept. The lack of unified coordination of EPPs does not contribute to the formation of Ukrainian students—future masters of adequate pharmaceutical practice competencies—and thus their acquisition of all ten current PRs, which are necessary for the activities of a pharmacist [28]. When considering the Model Standards of Practice for pharmaceutical specialists in Canada and the Pharmaceutical Education Standards of Great Britain, we also noticed an emphasis on the importance of the majority of PR pharmaceutical specialists, except entrepreneurs and agents of positive change [29,30].

Therefore, improving the image and clarification of the multifunctional work of pharmacists is the main task in terms of a clear distinction between PR and PF pharmacists, both in understanding pharmacists and patients.

It is appropriate when formulating EPP for pharmacists that the professional competencies and learning outcomes use a multifunctional model of difficult to organize, professional activities of a pharmacist, while simultaneously activating the PR of the pharmacist as an agent of positive change.

## 5. Conclusions

Detailing each pharmacist's 10 main PR into three or six corresponding partial PF resulted in a multifunctional model of difficult to organize, professional activities identified by a total of 50 PF. We note that the proposed definitions of a pharmacist's PF are open for further active discussion among pharmaceutical scientists and practitioners.

As a result, universities should use the proposed multifunctional model to formulate professional competencies and learner outcomes in EPP to prepare for a Master of Pharmacy. Furthermore, we believe that it is advisable to pay close attention to the development of multifunctional competencies of students—future masters for the quality of their implementation of all PR, and hence PF pharmacists in their future work in the pharmaceutical sector of health care.

**Author Contributions:** Conceptualization, Y.K., L.L. and B.H.; methodology, Y.K. and B.H.; investigation, Y.K., L.L., O.L. and B.H.; writing—review and editing, Y.K., L.L., R.L. and B.H.; visualization, Y.K., L.L. and B.H.; supervision, B.H.; project administration, R.L., O.L. and B.H. All authors have read and agreed to the published version of the manuscript.

**Funding:** The research received no external funding.

**Institutional Review Board Statement:** Not applicable.

**Informed Consent Statement:** Not applicable.

**Data Availability Statement:** Not applicable.

**Acknowledgments:** The authors would like to thank all the brave defenders of Ukraine who made the finalization of this article possible.

**Conflicts of Interest:** The authors declare no conflict of interest.

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
