# Peer review of "Detailing the Ten Main Professional Roles of a Pharmacist to Provide the Scope of Professional Functions"

_scipharm, doi:10.3390/scipharm91010005_

Round 1

Reviewer 1 Report

This manuscript details the professional roles of what is referred to as the “ten-star pharmacist” with respect to the professional functions of pharmacists. This detailing is clear, logical and likely very useful for educational programs of pharmacists. However, this is not an article—meaning that there is no study that was conducted to produce this detail. As such, it needs to be referred to as something other than an “Article”. The biggest problem of this submission is that the primary supporting research referenced for this analysis has been done by the same authors. There is no corresponding research in peer reviewed by other researchers to support the claims made in this paper regarding the “ten-star pharmacist” that has been presented, yet, the authors write as if their “ten-star pharmacist” is accepted and used worldwide. If this is true then they need to present the evidence that the “ten-star pharmacist” is known and used by pharmacists throughout the world. 

The English, though generally good, presents a number of problems that are repeated throughout the paper. The most numerous ones relate to the use of acronyms. Although the original phrases are plural, the authors again pluralize the acronyms. This must be corrected. The referencing is a more substantial difficulty. Not only are the references done in a style differing from MDPI requirements, there are too few references (this is especially so with respect to methodology) and the ones that are provided are often not published in peer reviewed journals. 

Once the authors present their paper as a viewpoint on how the “ten-star pharmacist” can be detailed, rather than as an article, and fix the problems that  have been mentioned and are further specified in the line by line suggested edits below, this paper likely will be a very useful tool for pharmacy education programs.

Line by line suggested edits

1 Given that this is not a scientific study but, rather, an analysis, the type of manuscript should be changed from “Article” to “Viewpoint” or to “Essay”.

2-3 Change “DETAILS OF THE MAIN PROFESSIONAL ROLES OF A PHARMACIST” to “Detailing the Ten Main Professional Roles of a Pharmacist to Provide the Scope of Professional Functions”

13-14 As the authors do not provide any discussion of the role of pharmacists in relation to either SARS-COV-2 or the armed attack of the Russian federation on Ukraine in the body of their work, please change “The pharmacist is the most accessible medical team member, which is well confirmed during the pandemic caused by SARS-CoV-2 and during the armed attack of the russian federation on Ukraine.” to “As members of a public trust profession, pharmacists are the most accessible medical team members.”. 

16 The term “decompose” is a technical term used in postmodern analysis. The authors are detailing the main professional roles of a pharmacist, they are not using postmodern analysis to decompose the term. As a result, change “decompose each pharmacist’s major” to “detail the major”.

15-17 Given that the authors have defined PR as professional roles and PF as professional functions, change “PRs into a pooled set of his PFs” to “PR into a pooled set of PF” (see suggested edit to lines 34-35 for why “his” has been removed).

18-27 Change all “PRs” to “PR” and “PRs” to “PR”.

30 Change “the “ten-star pharmacist”” to “”ten-star pharmacist””. Add “professional education programs”.

34 Please provide a reference to the meaning of “public trust profession”, such as https://doi.org/10.1515/ebce-2016-0001.

34-35 These days, academic articles avoid the use of male pronouns to describe universal conditions. Therefore, change “As a member of the public trust profession, the pharmacist plays an essential role in health care, as his work is “ to “As members of a public trust profession, pharmacists plays an essential role in health care, as their work is”

35-36 As mentioned regarding the Abstract, the authors do not provide any discussion of the role of pharmacists in relation to either SARS-COV-2 or the armed attack of the Russian federation on Ukraine, change “challenges, as evidenced by the SARS-CoV-2 pandemic and the armed attack of the russian federation on Ukraine.” to “challenges.”.

37 Although the meaning of the acronym “PR” is mentioned in the abstract, it needs to also be mentioned in the body of the paper before it is used. Therefore, change “the key PR of the pharmacist is to search for, make (production)” to “the key professional roles (PR) of the pharmacist are to search for, prepare, compound,”.

43 Change “PRs” to “PR”.

44 Change “the pharmacist's multifunctionality led to expanding his main PRs” to “pharmacists’ multifunctionality led to expanding their main PR”. Please provide a reference for this claim.

47 The authors claim that “the "ten-star pharmacist" concept is currently in force”. It needs to be made clear that this concept is one that has been specific to the Ukraine, Poland and, perhaps, Pakistan—it is not one that, according to the research, has been adopted internationally. To the extent that it has been adopted, this claim needs to be referenced. The authors’ current reference 16 provides a needed reference. However, as reference 16 is written by two of the same authors as this current paper, independent peer reviewed research is needed for considering that “the "ten-star pharmacist" concept is currently in force”. Change “PRs” to “PR”.

50 Change “PRs” to “PR”.

51 Some of the references cited are either not to the ten-star pharmacist concept or are not from peer-reviewed journals. The citations need to be to the 10-star pharmacist concepts and be to references in peer-reviewed journals.

55 Change “PRs” to “PR”.

56-57 Each of the methods mentioned must be referenced and a paragraph provided regarding what is involved in each method and why these methods were selected to provide this detailed analysis.

59 Change “PRs” to “PR”.

61 Change “detail” to “detailing”. Please provide a peer reviewed reference to the meaning of “principle of complexity limitation”.

62 Change “PFs” to “PF”.

63 Change “PRs” to “PR”.

68 The references provided for these claims are not to peer reviewed journals. Such references will need to be found for these claims to be made. Otherwise, the authors must detail why they have come to these conclusions.

71 Change “it” to “they”.

72 Change “PFs” to “PF”.

89 Change “PR2 is” to “PR2 are”

90 Change “PFs” to “PF”.

110 Change “PFs” to “PF”.

129-132 Change “The major PR4 of the pharmacist (a manager), according to which he/she must be able, based on professional knowledge of organizational management, to effectively influence employees and set up specific work, as well as positively perceive leadership from other professionals, is characterized by six partial PFs, namely” to “The major PR4 of the pharmacist as a manager include the ability, based on professional knowledge of organizational management, to effectively influence employees and set up specific work as well as positively perceive leadership from other professionals. They are characterized by six partial PF, namely”.

148-150 Change “Regarding the fifth main PR5 - a life-long-learner, according to which the pharmacist must begin with the stage of specialist training and continue training throughout his professional career, it is decomposed into three partial PFs, according to which the pharmacist” to “Regarding the fifth main PR5—being a life-long-learner, according to which pharmacists must begin with the stage of specialist training and continue training throughout their professional careers—it is detailed into three partial PF, according to which pharmacists”

155 Change “be an active participant” to “be active participants”.

159-161 Change “In turn, the basic PR6 (a teacher), according to which the pharmacist helps in the education and training of future generations of his colleagues and informing society, is divided into five partial PFs, i.e., the pharmacist must:” to “In turn, the basic PR6 (a teacher), where pharmacists help in the education and training of future generations of their colleagues and informing society, is divided into five partial PF, i.e., pharmacists must:”.

162 Change “to transfer” to “transfer”.

163 Change “to independent” to “to demonstrate independent”.

168-169 Change “protection, promotion, and prevention of health” to “protection and promotion of health”.

181 Change “this PR, the pharmacist implements five partial PFs, i.e., he/she” to “these PR, pharmacists implements five partial PF, i.e., they”.

204-205 Change “The pharmacist acts as a researcher as a result of the implementation of five partial PFs, during which he/she” to “Pharmacists act as researchers as a result of the implementation of five partial PF, during which they”

210 Change “to search for and develop new or improve existing drugs, to” to “search for and develop new or improve existing drugs”.

212 Change “to recognize” to “recognize”.

217 Change “properties” to “properties and that these will be effected by the individual size, weight, gender and age of consumers”.

230 Change “PR is” to “PR are”.

231 Change “PFs” to “PF”.

238 Change “to risk” to “risk”.

240 Change “to make” to “make”.

243 Change “to implement” to “implement”. Change “business to” to “business—to”.

249 Change “PFs” to “PF”.

262 Change “act” to “acting”.

269 Scientifically, this wasn’t a study. It was an analysis. Change "study" to "analysis" Change “PRs” to “PR”.

269-270 Change “decomposed” to “detailed”.

270 Change “verbal model of the multifunctional difficult to organized,” to “a multifunctional model of difficult to organize”.

271-275 Change each instance of “PFs” to “PF”.

276-278 Reference 14 is not to peer reviewed research. It is also inaccessible to a reader. This research, unless it has also been published in a peer reviewed journal, should not be cited.

278-284 If there is no peer reviewed research regarding the claims made in lines 276-278 then the rest of this paragraph must be rewritten to take this into consideration and not mention the research.

285 Change “Patient’s” to “The Patient’s”

286 Change “PRs” to “PR”.

287 Change “PFs” to “PF”. Change “his work” to “the pharmacist’s”.

299 Change “PRs” to “PR”. Change “PFs” to “PF”.

301-304 Change “Appropriate, on the one hand, when formulating in EPPs training of pharmacists special (professional) competencies and learning outcomes use a verbal model of multifunctional difficult to organized, professional activity a pharmacist, on the other hand -  activate PRs pharmacist as an agent of positive change.” to “It is appropriate when formulating EPP for pharmacists that the professional competencies and learning outcomes use a multifunctional model of difficult to organize professional activity of a pharmacist while simultaneously activating the PR of the pharmacist as an agent of positive change.”

306 Change “Decomposing” to “Detailing”. 

306-314 Change each instance of “PRs” to “PR” and of “PFs” to “PF”.

307 Change “verbal model of the multifunctional difficult to organized,” to “a multifunctional model of difficult to organize”.

309 Change “verbal model” to “multifunctional model”.

309-310 Change “special (professional)” to “professional”

310 Change “EPPs” to “EPP”.

References

Please refer to the MDPI style sheet regarding the correct method of referencing. All references must be redone in the correct style.

A number of the references are to not to publications in peer reviewed journals. Those that are not need to be substituted for ones that are found in peer reviewed journals.

Author Response

Dear reviewer!

We would like to thank You for revision and constructive comments that helped significantly improve the manuscript. Your suggestions have been incorporated in the revised manuscript (yellow highlight). 

We would like to comment the main points.

  • Given that this is not a scientific study but, rather, an analysis, the type of manuscript should be changed from “Article” to “Viewpoint” or to “Essay”.

Thanks for your suggestion. We agree. We asked the editor to change the type of article from "Article" to “Viewpoint”.

  • Change “DETAILS OF THE MAIN PROFESSIONAL ROLES OF A PHARMACIST” to “Detailing the Ten Main Professional Roles of a Pharmacist to Provide the Scope of Professional Functions”

Corrected as suggested.

  • 13-14 As the authors do not provide any discussion of the role of pharmacists in relation to either SARS-COV-2 or the armed attack of the Russian federation on Ukraine in the body of their work, please change “The pharmacist is the most accessible medical team member, which is well confirmed during the pandemic caused by SARS-CoV-2 and during the armed attack of the russian federation on Ukraine.” to “As members of a public trust profession, pharmacists are the most accessible medical team members.”. 

Corrected as suggested.

  • 16 The term “decompose” is a technical term used in postmodern analysis. The authors are detailing the main professional roles of a pharmacist, they are not using postmodern analysis to decompose the term. As a result, change “decompose each pharmacist’s major” to “detail the major”.

Corrected as suggested.

  • 15-17 Given that the authors have defined PR as professional roles and PF as professional functions, change “PRs into a pooled set of his PFs” to “PR into a pooled set of PF” (see suggested edit to lines 34-35 for why “his” has been removed).
  • 18-27 Change all “PRs” to “PR” and “PRs” to “PR”.
  • 30 Change “the “ten-star pharmacist”” to “”ten-star pharmacist””. Add “professional education programs”.

Corrected as suggested in all of the mentioned cases.

  • 34 Please provide a reference to the meaning of “public trust profession”, such as https://doi.org/10.1515/ebce-2016-0001.

The relevant reference [1] has been added.

  • 34-35 These days, academic articles avoid the use of male pronouns to describe universal conditions. Therefore, change “As a member of the public trust profession, the pharmacist plays an essential role in health care, as his work is “ to “As members of a public trust profession, pharmacists plays an essential role in health care, as their work is”
  • 35-36 As mentioned regarding the Abstract, the authors do not provide any discussion of the role of pharmacists in relation to either SARS-COV-2 or the armed attack of the Russian federation on Ukraine, change “challenges, as evidenced by the SARS-CoV-2 pandemic and the armed attack of the russian federation on Ukraine.” to “challenges.”.
  • Although the meaning of the acronym “PR” is mentioned in the abstract, it needs to also be mentioned in the body of the paper before it is used. Therefore, change “the key PR of the pharmacist is to search for, make (production)” to “the key professional roles (PR) of the pharmacist are to search for, prepare, compound,”.
  • 43 Change “PRs” to “PR”.

Corrected as suggested in all of the mentioned cases.

  • 44 Change “the pharmacist's multifunctionality led to expanding his main PRs” to “pharmacists’ multifunctionality led to expanding their main PR”. Please provide a reference for this claim.

Corrected as suggested and added references [4], [5], [6], according to the reference list

  • 47 The authors claim that “the "ten-star pharmacist" concept is currently in force”. It needs to be made clear that this concept is one that has been specific to the Ukraine, Poland and, perhaps, Pakistan—it is not one that, according to the research, has been adopted internationally. To the extent that it has been adopted, this claim needs to be referenced. The authors’ current reference 16 provides a needed reference. However, as reference 16 is written by two of the same authors as this current paper, independent peer reviewed research is needed for considering that “the "ten-star pharmacist" concept is currently in force”. Change “PRs” to “PR”.

Corrected as suggested and added references [7], [8], according to the reference list

  • 50 Change “PRs” to “PR”.
  • 55 Change “PRs” to “PR”.
  • 59 Change “PRs” to “PR”.
  • 62 Change “PFs” to “PF
  • 63 Change “PRs” to “PR”.
  • 71 Change “it” to “they”.
  • 72 Change “PFs” to “PF”.
  • 89 Change “PR2 is” to “PR2 are”
  • 90 Change “PFs” to “PF”.
  • 110 Change “PFs” to “PF”.
  • 155 Change “be an active participant” to “be active participants”.
  • 162 Change “to transfer” to “transfer”.
  • 163 Change “to independent” to “to demonstrate independent”.
  • 168-169 Change “protection, promotion, and prevention of health” to “protection and promotion of health”.
  • 212 Change “to recognize” to “recognize”.
  • 230 Change “PR is” to “PR are”.
  • 231 Change “PFs” to “PF”.
  • 238 Change “to risk” to “risk”.
  • 240 Change “to make” to “make”.
  • 243 Change “to implement” to “implement”. Change “business to” to “business—to”.
  • 249 Change “PFs” to “PF”.
  • 262 Change “act” to “acting”.
  • 269-270 Change “decomposed” to “detailed”.
  • 271-275 Change each instance of “PFs” to “PF”.
  • 285 Change “Patient’s” to “The Patient’s”
  • 286 Change “PRs” to “PR”.
  • 287 Change “PFs” to “PF”. Change “his work” to “the pharmacist’s”.
  • 299 Change “PRs” to “PR”. Change “PFs” to “PF”.
  • 306 Change “Decomposing” to “Detailing”. 
  • 306-314 Change each instance of “PRs” to “PR” and of “PFs” to “PF”.
  • 310 Change “EPPs” to “EPP”.

Corrected as suggested in all of the mentioned cases.

  • 51 Some of the references cited are either not to the ten-star pharmacist concept or are not from peer-reviewed journals. The citations need to be to the 10-star pharmacist concepts and be to references in peer-reviewed journals.

Suggested edits have been made. References [13], [14], [15], [16] were added.

  • 56-57 Each of the methods mentioned must be referenced and a paragraph provided regarding what is involved in each method and why these methods were selected to provide this detailed analysis.

Section 2 was expanded and detailed by adding the following references: [17], [18], [19] and [20].

  • 61 Change “detail” to “detailing”. Please provide a peer reviewed reference to the meaning of “principle of complexity limitation”.

Corrected as suggested.

  • 68 The references provided for these claims are not to peer reviewed journals. Such references will need to be found for these claims to be made. Otherwise, the authors must detail why they have come to these conclusions.

References were changed as suggested ([22] and [23], according to the revised list of references).

  • 129-132 Change “The major PR4 of the pharmacist (a manager), according to which he/she must be able, based on professional knowledge of organizational management, to effectively influence employees and set up specific work, as well as positively perceive leadership from other professionals, is characterized by six partial PFs, namely” to “The major PR4 of the pharmacist as a manager include the ability, based on professional knowledge of organizational management, to effectively influence employees and set up specific work as well as positively perceive leadership from other professionals. They are characterized by six partial PF, namely”.

Corrected as suggested.

  • 148-150 Change “Regarding the fifth main PR5 - a life-long-learner, according to which the pharmacist must begin with the stage of specialist training and continue training throughout his professional career, it is decomposed into three partial PFs, according to which the pharmacist” to “Regarding the fifth main PR5—being a life-long-learner, according to which pharmacists must begin with the stage of specialist training and continue training throughout their professional careers—it is detailed into three partial PF, according to which pharmacists”

Corrected as suggested.

159-161 Change “In turn, the basic PR6 (a teacher), according to which the pharmacist helps in the education and training of future generations of his colleagues and informing society, is divided into five partial PFs, i.e., the pharmacist must:” to “In turn, the basic PR6 (a teacher), where pharmacists help in the education and training of future generations of their colleagues and informing society, is divided into five partial PF, i.e., pharmacists must:”.

Corrected as suggested.

  • 181 Change “this PR, the pharmacist implements five partial PFs, i.e., he/she” to “these PR, pharmacists implements five partial PF, i.e., they”.

Corrected as suggested.

  • 204-205 Change “The pharmacist acts as a researcher as a result of the implementation of five partial PFs, during which he/she” to “Pharmacists act as researchers as a result of the implementation of five partial PF, during which they”

Corrected as suggested.

  • 210 Change “to search for and develop new or improve existing drugs, to” to “search for and develop new or improve existing drugs”.

Corrected as suggested.

  • 217 Change “properties” to “properties and that these will be effected by the individual size, weight, gender and age of consumers”.

Corrected as suggested.

  • 269 Scientifically, this wasn’t a study. It was an analysis. Change "study" to "analysis" Change “PRs” to “PR”.

Corrected as suggested.

  • 270 Change “verbal model of the multifunctional difficult to organized,” to “a multifunctional model of difficult to organize”.

Corrected as suggested.

  • 276-278 Reference 14 is not to peer reviewed research. It is also inaccessible to a reader. This research, unless it has also been published in a peer reviewed journal, should not be cited.

This reference has been removed and changed to reference [24] (revised list of references).

  • 278-284 If there is no peer reviewed research regarding the claims made in lines

Added reference [25] (revised list of references).

  • 301-304 Change “Appropriate, on the one hand, when formulating in EPPs training of pharmacists special (professional) competencies and learning outcomes use a verbal model of multifunctional difficult to organized, professional activity a pharmacist, on the other hand -  activate PRs pharmacist as an agent of positive change.” to “It is appropriate when formulating EPP for pharmacists that the professional competencies and learning outcomes use a multifunctional model of difficult to organize professional activity of a pharmacist while simultaneously activating the PR of the pharmacist as an agent of positive change.”

Corrected as suggested.

  • 307 Change “verbal model of the multifunctional difficult to organized,” to “a multifunctional model of difficult to organize”.

Corrected as suggested.

  • 309 Change “verbal model” to “multifunctional model”.

Corrected as suggested.

  • 309-310 Change “special (professional)” to “professional”

Corrected as suggested.

In addition, following Your comments, we also carefully revised the manuscript language, and hope the current version will be acceptable for publication.

Yours faithfully

Roman Lesyk

Reviewer 2 Report

Thank you for the opportunity to review this interesting paper which must have been put together in extremely challenging circumstances. This is an interesting paper fully deserving of further consideration, but one which I believe would receive far greater attention with some changes. 

Abbreviations should not be used, nor introduced in the abstract.  The full terms professional roles and professional functions should be used alone in the abstract and introduced in the introduction.

This work is fundamentally the authors' own interpretation of the expansion of the professional roles  into professional functions. Both the title and the text should reflect this; perhaps; 'Proposed deconstruction and expansion of the main professional roles of a pharmacist.' Whilst I feel sure that many will broadly agree with the proposed functions, essentially this is putting forward a proposal for broader peer-review and discussion. 

The methods section should be expanded; how were the proposed professional functions developed; by individual authors working in isolation? or by teams?  Were the proposed functions then passed to others in the group for review? 

The results section would benefit greatly by being divided into sections, with sub-headings, relating to each professional role. 

The discussion would benefit from including a comparison to the published professional standards and competencies for an entry level pharmacist from at least one country, and/or a published indicative curriculum for pharmacy degrees. How close do they come to the roles and functions proposed?  This would also make the conclusions more robust.

The conclusions should reflect that this proposed definition of functions is presented for broader review and consideration.

The team are to be congratulated for producing an excellent piece of work under the most challenging of circumstances.

Author Response

Dear reviewer!

We would like to thank You for revision and constructive comments that helped significantly improve the manuscript. Your suggestions have been incorporated in the revised manuscript (yellow highlight).

We would like to comment the main points.

  • Abbreviations should not be used, nor introduced in the abstract. The full terms professional roles and professional functions should be used alone in the abstract and introduced in the introduction.

Corrected as suggested.

  • This work is fundamentally the authors' own interpretation of the expansion of the professional roles into professional functions. Both the title and the text should reflect this; perhaps; 'Proposed deconstruction and expansion of the main professional roles of a pharmacist.' Whilst I feel sure that many will broadly agree with the proposed functions, essentially this is putting forward a proposal for broader peer-review and discussion.

The title of the manuscript was changed to "Detailing the ten main professional roles of a pharmacist to provide the scope of professional functions".

  • The methods section should be expanded; how were the proposed professional functions developed; by individual authors working in isolation? or by teams? Were the proposed functions then passed to others in the group for review?

Section 2 was expanded and detailed. References [17], [18], [19] and [20] (study design) have been added.

  • The results section would benefit greatly by being divided into sections, with sub-headings, relating to each professional role.

Sub-headings were added as suggested.

  • The discussion would benefit from including a comparison to the published professional standards and competencies for an entry level pharmacist from at least one country, and/or a published indicative curriculum for pharmacy degrees. How close do they come to the roles and functions proposed? This would also make the conclusions more robust.

The information on the comparison with the Canadian Professional Standard and the UK Pharmacy Education Standard was added, as well as references [27] and [28].

  • The conclusions should reflect that this proposed definition of functions is presented for broader review and consideration.

The necessary information is added to the conclusions.

In addition, we also carefully revised the manuscript language, and hope the current version will be acceptable for publication.

Yours faithfully

Roman Lesyk

Reviewer 3 Report

It was my pleasure to read a such valuable and important paper and I really appreciated its content. However, in my opinion, the manuscript can be improved in each of its structural parts.

The study aimed to decompose each of the major professional roles of pharmacists into a pooled set of his professional functions. 

The manuscript is written by specialists in management and economics working in connection with pharmaceutical education.

The method part of the manuscript needs a little expansion to include relevant data, information, knowledge and tools employed in the obtaining of the results. In my opinion some of these are already given in the results section and the authors needs only to revise a little the structure of the paper in order to accomplish the most part of the job. 

The mathematical and statistical tools employed should be introduced in the methods part. For instance, if some questionnaires and some grouping method has been used then their characteristics should be detailed.

Also the design of the study should be provided for the betterment understanding of the work taken as a whole.

Connection to similar papers, employing similar research methods and/or having similar study objectives may increase the value of the manuscript in the light of interdisciplinary research.

For instance in my view similar topics are: student perception of degree of academic community involvement and the role of the academic community in defining the professional route and the authors should strive themself to include relevant connections with papers discussing these topics.

Also evidence-based medicine self-assessment, knowledge, and integration into daily practice play an important role in connection with this study and should be identified the opportunity to construct a discussion about.

Author Response

Dear reviewer!

We would like to thank You for revision and constructive comments that helped significantly improve the manuscript. Your suggestions have been incorporated in the revised manuscript (yellow highlight). 

We would like to comment the main points.

  • The method part of the manuscript needs a little expansion to include relevant data, information, knowledge and tools employed in the obtaining of the results. In my opinion some of these are already given in the results section and the authors needs only to revise a little the structure of the paper in order to accomplish the most part of the job.

Section 2 was expanded and detailed. References [17], [18], [19] and [20] (study design) have been added.

  • The mathematical and statistical tools employed should be introduced in the methods part. For instance, if some questionnaires and some grouping method has been used then their characteristics should be detailed.

In the manuscript, we used for explanation and discussion data from a questionnaire survey from another study, therefore mathematical and statistical tools have not been given.

  • Also the design of the study should be provided for the betterment understanding of the work taken as a whole. Connection to similar papers, employing similar research methods and/or having similar study objectives may increase the value of the manuscript in the light of interdisciplinary research.

The information on the comparison with the Canadian Professional Standard and the UK Pharmacy Education Standard was added, as well as references [27] and [28].

  • For instance in my view similar topics are: student perception of degree of academic community involvement and the role of the academic community in defining the professional route and the authors should strive themself to include relevant connections with papers discussing these topics.

Thank you very much for your recommendations regarding the study of students' perception of the degree of involvement of the academic community and the role of the academic community in determining the career path. In our manuscript, we did not aim to investigate this aspect, but we believe that it is compatible with the field of our research and we will certainly take it into account in further research.

  • Also evidence-based medicine self-assessment, knowledge, and integration into daily practice play an important role in connection with this study and should be identified the opportunity to construct a discussion about.

Sentence regarding that: "the proposed definitions of PF pharmacists are open for further active discussion among pharmaceutical scientists and practitioners" is added to the conclusions of the manuscript.

In addition, we also carefully revised the manuscript language, and hope the current version will be acceptable for publication.

Yours faithfully

Roman Lesyk

Round 2

Reviewer 1 Report

Thank you to the authors for making the suggested edits to the manuscript. It is much improved as a result, and that much closer to publication. 

There remain some points, thought corrected, were not changed in a way that was not as expected by this reviewer. As such, in these instances, some problems still remain. They are detailed below in the line by line suggested edits.

Line by line suggested edits

1 Change “Article” to “Viewpoint”.

15 Change “role” to “roles”.

16 Change “set of his PF” to “set of PF”.

22 Change “which allowed to obtain from three to six partial PFs for the respective PR” to “which allowed three to six partial PF for the respective PR to be obtained”.

46 In the previous review by this reviewer, the following point was made “The authors claim that “the "ten-star pharmacist" concept is currently in force”. It needs to be made clear that this concept is one that has been specific to the Ukraine, Poland and, perhaps, Pakistan—it is not one that, according to the research, has been adopted internationally. To the extent that it has been adopted, this claim needs to be referenced.” Although the authors now have two new references, neither indicates that the “ten-start pharmacist” concept is currently in force. Unless these authors can find a reference demonstrating that this concept is in force internationally, this statement must be dropped from this manuscript.

56-63 Thank you for adding this clarifying and referenced material. This reviewer should have been clearer in saying that the references should be ones from within the last five years. As such, references 17-20 are too old to be appropriate for demonstrating that these methods are considered the standard today. Please find current references.

73 Change “detailed PFs of the main PRs” to “detailed PF of the main PR”.

75 Thank you for changing the numbering of the sections; however, the number also requires a title. A suggestion is “3.1. Provider”. This (and the other changes to follow for section 3) should be a separate title from the body of the paragraph.

93 Change “3.2.” to “3.2. Decision maker”.

114 Change “3.3.” to “3.3. Communicator”.

134 Change “3.4” to “3.4. Manager”.

153 Change “3.5” to “3.5. Life-long learner”.

164 Change “3.6” to “3.6. Teacher”.

182 Change “3.7” to “3.7. Leader”.

206 Change “3.8” to “3.8. Researcher”.

233 Change “3.9” to “3.9. Entrepreneur”.

252 Change “3.10. to “3.10. Agent of positive change”.

254 The author is reminded that PR is plural. As a result, change “We described this PR” to “This final one of the PR is described”.

275 Change “ten main” to “the ten main”.

291 Change “Patient’s” to “patient’s”.

293-294 Change “pharmacist’s” to “pharmacist”.

294 Reference 25 is outdated. Please find a current reference in a peer reviewed journal to support this claim.

316 Change “PFs” to “PF”.

318 Change “PF pharmacists” to “pharmacist’s PF”.

321 Change “learn outcomes” to “learner outcomes”.

Author Response

Dear reviewer!

We would like to thank You for revision and constructive comments that helped significantly improve the manuscript. Your suggestions have been incorporated in the revised manuscript (yellow highlight).

We would like to comment the main points.

  • 1 Change “Article” to “Viewpoint”.
  • 15 Change “role” to “roles”.
  • 16 Change “set of his PF” to “set of PF”.
  • 22 Change “which allowed to obtain from three to six partial PFs for the respective PR” to “which allowed three to six partial PF for the respective PR to be obtained”.

Corrected as suggested in all of the mentioned cases.

  • 46 In the previous review by this reviewer, the following point was made “The authors claim that “the "ten-star pharmacist" concept is currently in force”. It needs to be made clear that this concept is one that has been specific to the Ukraine, Poland and, perhaps, Pakistan—it is not one that, according to the research, has been adopted internationally. To the extent that it has been adopted, this claim needs to be referenced.” Although the authors now have two new references, neither indicates that the “ten-start pharmacist” concept is currently in force. Unless these authors can find a reference demonstrating that this concept is in force internationally, this statement must be dropped from this manuscript.

Thank you very much for the important remark. In the manuscript, we specified that the "seven-star pharmacist" concept was accepted at the international level in 1997. In addition, appropriate corrections were made, namely, the interpretation was changed to: "So, as of now, the "ten-star pharmacist" concept is spreading in the world. This concept is gaining scientific justification and content at the level of researchers from different countries ". Also added references: 10 and 11, namely:

  1. Mattingly, T.J. 2nd.; Mullins, C.D.; Melendez, D.R., Boyden, K., Eddington, N.D. A Systematic Review of Entrepreneurship in Pharmacy Practice and Education. Am. J. Pharm. Educ., 2019, 83, 3, 7233. DOI: https://doi.org/10.5688/ajpe7233.
  2. Hallit, S.; Hajj, A.; Sacre, H.; Zeenny, R.M.; Akel, M.; Sili, G.; Salameh, P. Emphasizing the Role of Pharmacist as a Researcher: The Lebanese Order of Pharmacists' Perspective. J. Res. Pharm. Pract., 2019, 27, 8, 4, 229-230. DOI: https://doi.org/10.4103/jrpp.JRPP_19_7.

  • 56-63 Thank you for adding this clarifying and referenced material. This reviewer should have been clearer in saying that the references should be ones from within the last five years. As such, references 17-20 are too old to be appropriate for demonstrating that these methods are considered the standard today. Please find current references.

Rererence 17 is replaced by 19 in the new version of the bibliography, namely:

  1. Vears, D.F.; Gillam, L. Inductive content analysis: A guide for beginning qualitative researchers. Health Prof. Educ. 2022, 23, 1, 111–127. DOI: https://doi.org/10.11157/fohpe.v23i1.544.

The authors request to leave references18, 19, 20 (in the new version of the reference list 20, 21, 22), since at the request of the first review they contain references to the essence of well-known methods used in the manuscript. Later publications in peer-reviewed journals provide a general definition of these methods without characterizing their essence.

  • 73 Change “detailed PFs of the main PRs” to “detailed PF of the main PR”.

Corrected as suggested.

  • 75 Thank you for changing the numbering of the sections; however, the number also requires a title. A suggestion is “3.1. Provider”. This (and the other changes to follow for section 3) should be a separate title from the body of the paragraph.
  • 75 Change “3.1.” to “3.1. Caregiver”.
  • 93 Change “3.2.” to “3.2. Decision maker”.
  • 114 Change “3.3.” to “3.3. Communicator”.
  • 134 Change “3.4” to “3.4. Manager”.
  • 153 Change “3.5” to “3.5. Life-long learner”.
  • 164 Change “3.6” to “3.6. Teacher”.
  • 182 Change “3.7” to “3.7. Leader”.
  • 206 Change “3.8” to “3.8. Researcher”.
  • 233 Change “3.9” to “3.9. Entrepreneur”.
  • 252 Change “3.10. to “3.10. Agent of positive change”.

Corrected as suggested in all of the mentioned cases.

  • 254 The author is reminded that PR is plural. As a result, change “We described this PR” to “This final one of the PR is described”.
  • 275 Change “ten main” to “the ten main”.
  • 291 Change “Patient’s” to “patient’s”.
  • 293-294 Change “pharmacist’s” to “pharmacist”.

Corrected as suggested in all of the mentioned cases.

  • 294 Reference 25 is outdated. Please find a current reference in a peer reviewed journal to support this claim.

Reference 25 is changed to 26 and 27, namely:

  1. Guhl, D.; Blankart, K.E.; Stargardt, T. Service quality and perceived customer value in community pharmacies. HSMR. 2019, 32, 1, 36–48. DOI: https://doi.org/10.1177/0951484818761730
  2. Newlands, R.S. ; Power, A. ; Young, L. ; Watson, M. Quality improvement of community pharmacy services: a prioritisation exercise. Int. J Pharm. Pract. 2018, 26, 1, 39-48. DOI: https://doi.org/10.1111/ijpp.12354.

  • 316 Change “PFs” to “PF”.
  • 318 Change “PF pharmacists” to “pharmacist’s PF”.
  • 321 Change “learn outcomes” to “learner outcomes”.

Corrected as suggested in all of the mentioned cases.

Yours faithfully

Roman Lesyk

Reviewer 2 Report

My thanks to the authors. I believe my concerns have been adequately addressed and now welcome publication

Author Response

Dear reviewer!

We would like to thank You for revision and constructive comments that helped significantly improve the manuscript. We appreciate your positive assessment of our work.

Yours faithfully

Roman Lesyk